# Patient-derived xenograft culture-transplant system for investigation of human breast cancer metastasis

Dennis Ma[1,7], Grace A. Hernandez[2,7], Austin E. Y. T. Lefebvre[3], Hamad Alshetaiwi[1,4], Kerrigan Blake[5], Kushal R. Dave[2], Maha Rauf[1], Justice W. Williams[1], Ryan T. Davis [2], Katrina T. Evans[2], Aaron Longworth [2], Madona Y. G. Masoud[2], Regis Lee[2], Robert A. Edwards[6], Michelle A. Digman [3], Kai Kessenbrock [1] & Devon A. Lawson [2✉]

Metastasis is a fatal disease where research progress has been hindered by a lack of authentic experimental models. Here, we develop a 3D tumor sphere culture-transplant system that facilitates the growth and engineering of patient-derived xenograft (PDX) tumor cells for functional metastasis assays in vivo. Orthotopic transplantation and RNA sequencing (RNA-seq) analyses show that PDX tumor spheres maintain tumorigenic potential, and the molecular marker and global transcriptome signatures of native tumor cells. Tumor spheres display robust capacity for lentiviral engineering and dissemination in spontaneous and experimental metastasis assays in vivo. Inhibition of pathways previously reported to attenuate metastasis also inhibit metastasis after sphere culture, validating our approach for authentic investigations of metastasis. Finally, we demonstrate a new role for the metabolic enzyme NME1 in promoting breast cancer metastasis, providing proof-of-principle that our culture-transplant system can be used for authentic propagation and engineering of patient tumor cells for functional studies of metastasis.

[1] Department of Biological Chemistry, University of California, Irvine, CA, USA. [2] Department of Physiology and Biophysics, University of California, Irvine, CA, USA. [3] Department of Biomedical Engineering, University of California, Irvine, CA, USA. [4] Department of Pathology, University of Hail, Hail, Saudi Arabia. [5] Center for Complex Biological Systems, University of California, Irvine, CA, USA. [6] Department of Pathology & Laboratory Medicine, University of California, Irvine, CA, USA. [7] These authors contributed equally: Dennis Ma, Grace A. Hernandez. ✉email: dalawson@uci.edu

Metastasis is the cause of >95% of breast cancer patient mortality, causing death in >40,000 women per year[1,2]. There are limited effective treatments for metastasis, and the identification of new therapeutic targets has been hindered by a lack of experimental models that faithfully recapitulate metastatic disease. Most metastasis research has been conducted with a handful of genetically engineered mouse models and cell lines that have limitations for accurately modeling metastasis in humans[3,4]. Patient-derived xenograft (PDX) models, where patient tumors are propagated in mice, offer increased authenticity as they maintain intratumoral heterogeneity, molecular markers, and pathological characteristics of the original patient tumor[5–7]. PDX models have been used extensively in pre-clinical studies for drug testing in vivo, but their use in metastasis research has been limited by technical challenges. Functional studies of metastasis often involve genetic or pharmacologic perturbation of cancer cells in culture followed by transplantation in vivo to determine the effect of a gene or pathway of interest on metastatic dissemination. Although studies have reported transient cultures of PDX cells for drug testing in vitro[8–10], they are typically only maintained for 24–72 h and not used for viral engineering or transplantation. The development of a robust method for the propagation, engineering, and transplantation of PDX tumor cells would enable new functional investigations of metastasis using patient tumor cells.

Although human cancer cell lines are typically maintained in 2D culture, previous work has shown that 3D cultures provide several advantages for cultivating primary tumor cells. As they are 3D, they more closely recapitulate the physiologic conditions found in normal tissues where cell phenotype and function are heavily influenced by cell–cell and cell-extracellular matrix (ECM) interactions[11]. 3D conditions can be generated using various conditions, such as the hanging drop method used to maintain embryonic stem (ES) cells[12–14], suspension in low-adhesion conditions as used for neurosphere cultures[15], or embedding in ECM scaffolds such as collagens, matrigel, or synthetic hydrogels, which have been used to enrich for cancer stem cells[16]. Here, we screened several 3D culture conditions and present an optimized method for the propagation of PDX cells as tumor spheres. RNA sequencing (RNA-seq) analysis shows PDX sphere cells maintain gene expression signatures characteristic of uncultured PDX tumor cells. Sphere cells can be engineered with lentivirus and produce tumors that recapitulate native PDX tumors following orthotopic transplantation in vivo. Importantly, we find that PDX sphere cells yield robust spontaneous metastasis from orthotopic tumors, as well as experimental metastasis following intravenous (i.v.) and intracardiac (i.c.) delivery in vivo. Inhibition of known metastasis-promoting pathways attenuates metastasis following tumor sphere culture, validating our approach for authentic investigations of metastasis from patient tumor cells. Finally, we investigate the role of nucleoside diphosphate kinase A (NME1), a gene we previously found upregulated during metastatic seeding of PDX tumor cells[17]. We find that NME1 overexpression promotes metastasis of orthotopically transplanted, cultured PDX cells, providing proof-of-principal for the value of this culture-transplant system for facilitating functional analyses of new genes of interest in patient tumor cells.

## Results

### Development of a 3D culture system for viable propagation of PDX cells as tumor spheres in vitro. We screened several culture conditions to develop an optimal method for viable propagation of PDX tumor cells in vitro (Fig. 1a). We assessed two 3D growth methods: (1) suspension, where cells are grown in ultra-low attachment (ULA) plates[18,19], and (2) matrigel, where cells are grown in a basement membrane-rich semi-solid substratum[20] (Fig. 1a). We also tested two media conditions: (1) mammary epithelial cell growth medium (MEGM), which supports short-term cultures of PDX cells for drug testing[8], and (2) EpiCult™-B (EpiCult), which is commonly used for breast epithelial cell culture[21–25] (Fig. 1a).

We used the previously established breast cancer PDX models, HCI001, HCI002, HCI010 (ER−PR−Her2−; basal-like), and HCI011 (ER+PR+Her2−, luminal B)[5] (Fig. 1a). We screened the culture conditions using HCI010 and then validated the results using the other models. HCI010 tumors were harvested from PDX mice after 2–5 months of growth in vivo, digested to single-cell suspensions, and plated into four culture conditions: (i) ULA plates and EpiCult (ULA-E); (ii) ULA plates and MEGM (ULA-M); (iii) matrigel and EpiCult (MAT-E), and (iv) matrigel with MEGM (MAT-M) (Fig. 1a). Microscopy 7–14 days later showed ULA-E and MAT-E both produced spheroid structures, while ULA-M and MAT-M produced limited growth (Fig. 1b). Flow cytometry analysis of cell viability by annexin V (aV) and propidium iodide (PI) staining showed that MAT-E produced the highest percentage of aV−PI− viable cells (60.6 ± 2.0%; $p < 0.0001$) (Fig. 1c, Supplementary Fig. 1a). ULA-E showed 2.3-fold ($p < 0.0001$) lower viability than MAT-E and 1.4-fold ($p < 0.0001$) more aV+PI− cells in early apoptosis (Fig. 1c, Supplementary Fig. 1a), indicating MAT-E is superior for producing viable PDX sphere growth.

We next compared the culture conditions for their ability to expand and passage PDX cells. Live-cell number was determined before plating and after 9 days of culture. Cell counts showed that MAT-E produced a 2.2-fold cell increase in cell number, which was greater than ULA-E ($p = 0.014$), and the other conditions produced a minimal increase or decrease in cell number (Fig. 1d). Passaging experiments showed MAT-E could also support extended viable growth of PDX cells. First-generation (P0) spheres were harvested, dissociated, and re-plated for an additional 14–17 days to produce P1 spheres (Fig. 1a). Flow cytometry analysis of P1 cells showed that MAT-E produced fourfold greater viability than MAT-M ($p < 0.0001$) (Fig. 1e). Cell counts showed that MAT-E produced a twofold increase of viable cells, which was greater than MAT-M ($p = 0.008$) that reduced total cell number (Fig. 1f).

Further evaluation of MAT-E showed it is sufficient to support the growth of other PDX models. HCI001, HCI002, and HCI011 tumors were harvested from PDX mice, dissociated to single-cell suspensions, and plated in MAT-E conditions. Microscopy analysis showed a clear outgrowth of spheres from all three models (Supplementary Fig. 1b). Passaging experiments for HCI002 showed a similar cell number increase in MAT-E conditions as observed for HCI010 (Supplementary Fig. 1c). We also confirmed that spheres were generated from PDX tumor cells and not contaminating mouse epithelium using a flow cytometry assay previously developed by our laboratory[26]. Spheres generated from each PDX model were dissociated and stained with species-specific antibodies for CD298 (human) and MHC-I (mouse). Flow cytometry analysis showed that >90% of cells were CD298+MHC-I− human tumor cells (Supplementary Fig. 1d). These data show that MAT-E is superior for the viable growth of human PDX tumor cells in vitro. Supplementary Fig. 1e and Supplementary movies 1–3 show the kinetics of sphere growth in MAT-E conditions using time-lapse imaging.

### PDX tumor sphere cells maintain their global transcriptome program. To determine whether PDX cells maintain their native state following MAT-E culture, we compared the global transcriptome profiles of cultured and uncultured cells by RNA-seq

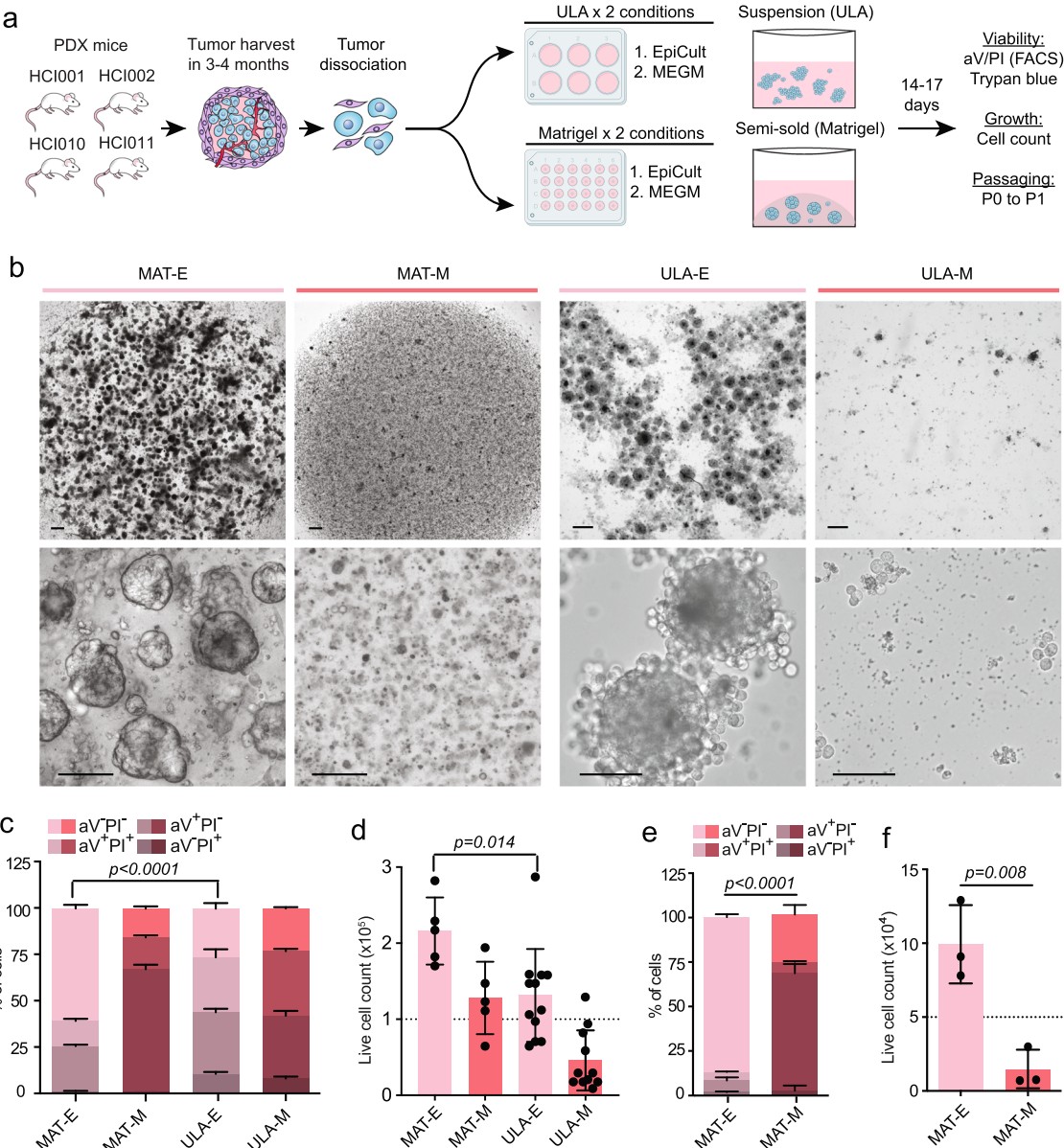

**Fig. 1 Comparison of 3D culture methods for viable propagation of PDX tumor cells in vitro. a** Schematic summarizes culture conditions compared and readouts for PDX cell viability, growth, and passaging capacity. **b** Representative brightfield images show spheroid structures generated 7–14 days after plating $2.5 \times 10^5$ HCI010 cells in MAT-E, MAT-M, ULA-E, and ULA-M conditions. Scale bars = 300 μm (top row), 100 μm (bottom row). **c** Flow cytometry analysis of HCI010 cell viability post culture by aV and PI staining. The bar graph shows the percent of aV⁻PI⁻ (live), early apoptotic (aV⁺PI⁻), and dead (aV⁺PI⁺, aV⁻PI⁺) HCI010 cells in each condition. *P* value determined by unpaired *t* test comparing the percentage of aV⁻PI⁻ viable cells in MAT-E vs ULA-E, $n = 3$ per condition. Data represented as mean ± s.d. **d** Bar graph shows total viable HCI010 cell number in each condition at the conclusion of culture. Cells were plated at $1.0 \times 10^5$ cells/well (dashed line) and quantified 9 days later by trypan blue exclusion. *P* value determined by unpaired *t* test comparing the number of live cells in MAT-E vs ULA-E, $n = 5$–11 wells per condition. Data represented as mean ± s.d. **e** Flow cytometry analysis of HCI010 cell viability in each condition post passaging by aV and PI staining. HCI010 cells were cultured for 2 weeks, dissociated, re-plated at $5.0 \times 10^4$ cells/well in MAT-E and MAT-M, and analyzed 21 days later. *P* value determined by unpaired *t* test comparing aV⁻PI⁻ viable cells, $n = 3$. Data represented as mean ± s.d. **f** Bar graph shows total viable HCI010 cell number in each condition after passaging as described in **e**. Cell number was determined by trypan blue exclusion, and the dashed line indicates the initial cell number plated. *P* value determined by unpaired *t* test, n = 3. Data represented as mean ± s.d.

(Fig. 2a). Cells from HCI002 ($n = 3$) and HCI010 ($n = 3$) tumors were dissociated and split into two matched groups, one for culture in MAT-E (cultured 1–3) and the other for immediate library preparation (uncultured 1–3) (Fig. 2a). CD298⁺MHC-I⁻ cells from both groups were sorted by flow cytometry to enrich human tumor cells and sequenced at 35 million paired-end reads per sample. Reads were aligned to the human hg38 reference genome, and differentially expressed genes were identified using *DESeq2*[27].

Gene expression was compared in paired cultured and uncultured samples from each PDX model by Pearson's correlation test (Fig. 2b). This showed that gene expression was strongly correlated in most pairs, indicating that culture does not substantially alter cell state (Fig. 2b). Differential expression analysis identified 1732 genes up and downregulated following MAT-E culture that was conserved across all pairs (logFC > 2.0, $p < 0.05$) (Fig. 2c, Supplementary Data 1). This represents only 7.7% of the total transcriptome, further indicating that culture

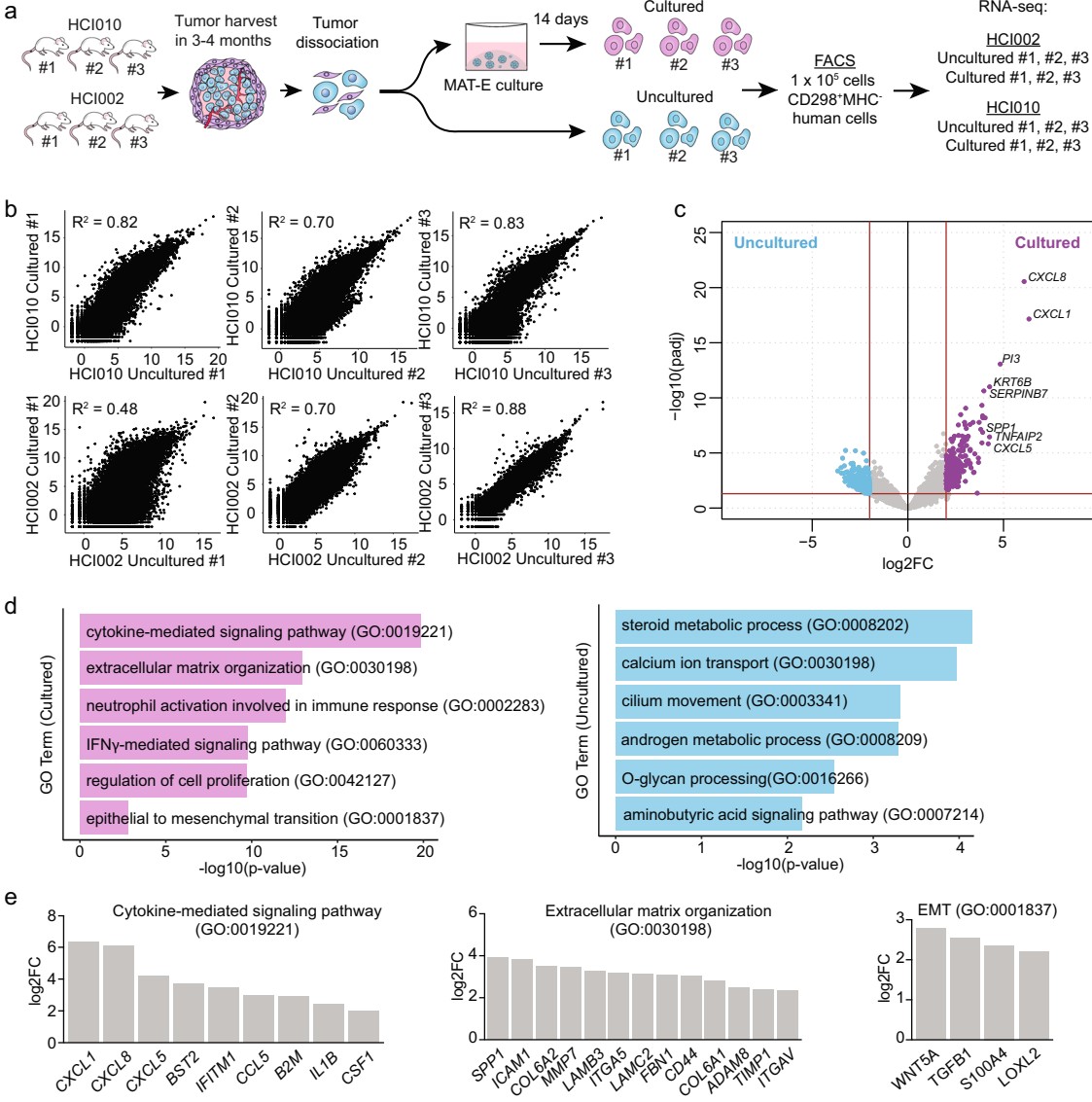

**Fig. 2 PDX tumor sphere cells maintain their global transcriptome program. a** Workflow schematic for generation of RNA-seq data set to compare transcriptome signatures in paired cultured and uncultured cells from HCI002 and HCI010. **b** Scatter plots show the correlation of gene expression across the entire transcriptome in paired cultured and uncultured samples (n = 22,446 genes). Pearson's correlation coefficient ($R^2$) is shown for each pair. **c** Volcano plot shows genes significantly upregulated in cultured (n = 277 genes) or uncultured (n = 1455 genes) cells that are conserved in all pairs (log2FC > 2, adj p < 0.05). See Supplementary Data 1 for the full gene list. FC = Fold change. Adjusted P value determined in DESeq2 by Benjamini–Hochberg adjustment of Wald test P alues. **d** Bar graph shows GO terms associated with genes upregulated in cultured (n = 277 genes) and uncultured (n = 1455 genes) cells (log2FC > 2, adj p < 0.05). P value determined using Enrichr by Fisher's exact test. **e** Bar graphs show select GO term-associated genes upregulated in cultured cells. See Supplementary Data 1 for the full gene list. FC = Fold change. P value determined by Wald test.

does not substantially alter cell state. Notably, canonical breast cell differentiation genes and molecular markers (*KRT8*, *KRT18*, *KRT14*, *TP63*, *ERBB2*, *ESR1*, *PGR*) were not differentially expressed, suggesting the cells maintain their differentiation state and tumor subtype (Supplementary Data 1).

Gene ontology analysis of the differentially expressed genes revealed several pathways up and downregulated following MAT-E culture (Fig. 2d). Pathways significantly upregulated include "cytokine-mediated signaling pathway", "extracellular matrix (ECM) organization", "interferon-gamma (IFNγ)-mediated signaling pathway", "regulation of cell proliferation", and "epithelial to mesenchymal (EMT) transition" (Fig. 2d). Analysis of the genes within each GO term identified numerous chemokines (*CXCL1*, *CXCL8*, *CXCL5*), cytokines (*CSF1*, *CCL5*, *IL1β*), and IFN-response genes (*BST2*, *B2M*, *HLA*), suggesting that culture induces a stress-

related or inflammatory response (Fig. 2e). We also observed upregulation of 23 ECM-related genes, including *ICAM1*, *COL6A2*, *MMP7*, *ITGA5*, and *LAMC2* (Fig. 2e). Interestingly, cultured cells also displayed increased expression of the EMT-associated genes *WNT5A*, *TGFB1*, *S100A4*, and *LOXL2*, as well as the CSC marker *CD44* (Fig. 2e, Supplementary Data 1). Prior work has shown that normal and breast cancer stem cells display EMT features[28,29]. This may therefore indicate that MAT-E culture selects for stem-like cells, which is consistent with prior reports using 3D culture conditions[30,31]. However, it is also possible that MAT-E culture induces upregulation of these transcriptome programs.

**PDX tumor sphere cells maintain tumorigenic potential and form spontaneous metastasis in vivo.** Spontaneous metastasis models are often considered more authentic than experimental

metastasis models since the cells must progress through each step of the metastatic cascade in order to metastasize[32]. Prior work has shown that HCI002 and HCI010 develop basal-like primary tumors and spontaneous metastasis following orthotopic transplantation[5,26]. We investigated whether PDX cells maintain these capabilities following sphere culture. We cultured and expanded HCI002 and HCI010 cells in MAT-E for 2 weeks and orthotopically transplanted serial dilutions of cells into NOD/SCID mice (Fig. 3a). Similar to prior reports with uncultured PDX cell transplants[26], we observed primary tumor growth from ~100% of animals transplanted with cultured cells at each dilution (Fig. 3b, Supplementary Fig. 2a).

We next compared histopathological and molecular features of MAT-E-cultured and native PDX tumors (Fig. 3c, Supplementary Fig. 2b). All HCI010 tumors were poorly vascularized, necrotic, and showed no epithelioid differentiation, with little nodularity or ductal structure (Fig. 3c). HCI002 tumors showed scant intervening myxoid stroma and thin-walled vessels, less necrosis, and retained some small areas of nodular aggregates but were mostly undifferentiated (Supplementary Fig. 2b). We observed no pathological differences between native and cultured tumors. Immunofluorescence (IF) staining for canonical basal (KRT5), myoepithelial (SMA), and luminal (KRT18) cell markers further indicated that HCI010 MAT-E tumors maintain their native differentiation state (Fig. 3c). Tumors from both conditions displayed widespread KRT5 and KRT18 expression, and limited SMA expression, as expected for basal-like tumors (Fig. 3c).

We further evaluated the capacity of orthotopic tumors to produce spontaneous metastasis. Flow cytometry analysis identified CD298+MHC-I− human metastatic cells in the lungs (11/14), lymph nodes (5/14), bone marrow (2/5), and peripheral blood (2/5) but not the brains (0/5) of transplanted animals (Fig. 3d). These frequencies were consistent with prior reports of spontaneous metastasis from uncultured PDX cells (Supplementary Fig. 2c)[26]. However, the metastatic burden was low in most tissues, so we tested the capacity of cultured cells to produce spontaneous metastasis in NOD scid gamma (NSG) mice, which lack natural killer (NK) cells and support greater human cell engraftment[26]. Remarkably, NSG mice displayed >32-fold higher metastatic burden in the lungs than NOD/SCID mice ($p < 0.0001$) (Fig. 3e), and metastatic lesions were detectable by histopathological analysis (Fig. 3f). This is consistent with reports demonstrating the importance of NK cells in controlling metastasis and shows that different immunodeficient strains can be used to achieve specific experimental contexts[33]. Importantly, we also observed spontaneous metastasis from HCI002 transplants, which displayed lung metastasis (5/5 mice) and circulating tumor cells in the peripheral blood (3/5 mice) (Supplementary Fig. 2d, e). These data show that PDX cells maintain their tumorigenic and metastatic potential after sphere culture and highlight the utility of our culture-transplant system for studies of spontaneous metastasis from human patient cells.

**PDX tumor sphere cells can be genetically engineered for functional studies of metastasis in vivo.** The ability to genetically perturb genes of interest is critical for functional investigations of metastasis. We investigated whether PDX tumor sphere cells can be genetically engineered and transplanted to generate spontaneous metastasis in vivo. Using a GFP lentiviral construct, we tested the infection efficiency of HCI002 and HCI010 sphere cells grown in MAT-E conditions (Fig. 4a). PDX cells were grown for 2–3 weeks to generate P0 spheres and then harvested and dissociated to generate cell suspensions. P0 cell suspensions were transduced using a combined centrifugation and suspension infection protocol (see Methods), and re-plated in MAT-E

conditions for 2–3 weeks to expand and generate P1 spheres (Fig. 4a). Fluorescence microscopy revealed robust GFP expression both within and between P1 spheres (Fig. 4b, c). Quantification of infection efficiency by flow cytometry showed successful transduction with 75.2 ± 6.4% and 23.2 ± 7.6% of HCI010 and HCI002 cells positive for GFP, respectively (Fig. 4d). We next determined whether transduced P1 sphere cells could retain GFP signal and produce spontaneous metastasis following orthotopic transplantation in vivo. Flow cytometry sorted GFP+ HCI002 and HCI010 P1 sphere cells were transplanted, and primary tumors and lungs were collected and analyzed 60 days later. Whole-mount imaging showed that GFP expression was retained in primary tumors (Fig. 4e), and quantification by flow cytometry showed that >90% of CD298+MHC-I− human PDX tumor cells retained GFP signal (Fig. 4f). Analysis of lung tissue sections showed the presence of GFP-positive metastatic lesions (Fig. 4g), and flow cytometry analysis of lung cell suspensions identified CD298+MHC-I−GFP+ human metastatic cells (Fig. 4h). Remarkably, >95% of CD298+MHC-I− human metastatic cells in the lung also retained GFP expression (Fig. 4h). Thus, we demonstrate PDX sphere cells can be robustly engineered, maintain their capacity to form primary tumors and spontaneous metastasis and retain their genetic alterations in vivo.

**PDX tumor sphere cells produce robust experimental metastasis in vivo.** In experimental metastasis models, cells are injected directly into the circulation via i.v. or i.c. delivery. I.v. injection introduces cells into the venous circulation and favors lung metastasis, whereas i.c. injection delivers cells into arterial circulation and supports bone and brain metastasis[17,34,35]. These approaches produce metastasis with more robust, rapid, and reproducible kinetics than spontaneous models and enable direct investigation of later steps in the metastatic cascade, such as survival in the bloodstream, extravasation, and seeding in the distal tissue. We investigated whether PDX cells expanded in culture will metastasize following i.v. or i.c. delivery in vivo.

We harvested HCI010 and HCI002 tumors from PDX mice and cultured the cells for two weeks in MAT-E conditions (Fig. 5a). In all, $5 \times 10^5$ P0 sphere cells were injected i.v. or i.c. into NOD/SCID mice and peripheral tissues were harvested and analyzed by flow cytometry and histology eight weeks later (Fig. 5a). In i.c.-injected animals, we observed CD298+MHC-I− human HCI010 cells in the lungs (4/5 mice), bone marrow (3/5 mice), brain (5/5 mice), and peripheral blood (5/5 mice), but not the lymph nodes (0/5 mice) (Fig. 5b). Metastatic burden was remarkably robust in the brain of some animals, where HCI010 cells constituted >30% of live cells (Fig. 5b). However, substantial variation was observed, where other animals showed lower metastatic burden (1% of live cells) (Fig. 5b). Histopathological analysis confirmed the presence of large metastatic lesions in the brains, and IF staining showed specific expression of the basal cancer subtype marker KRT5 as well as the proliferation marker Ki67 (Fig. 5c, d, Supplementary Fig. 3a). We also observed metastasis in the lungs, brain and peripheral blood of animals injected i.c. with HCI002 cells (Supplementary Fig. 3b). Like HCI010, metastasis was most robust in the brain, demonstrating the utility of our culture-transplant system for studies of brain metastasis using patient tumor cells.

In i.v.-injected animals, we observed CD298+MHC-I− HCI010 human cells in the lungs (5/5 mice), bone marrow (1/5 mice), and peripheral blood (1/5 mice), but not the lymph nodes (0/5 mice) (Fig. 5e). As expected, i.v. delivery yielded the highest percent and burden in the lungs (0.96 ± 0.48%) (Fig. 5e). Analysis of mice injected i.v. with HCI002 cells showed limited metastasis (Supplementary Fig. 3c). We subsequently tested whether i.v. injection into NSG mice would support greater metastasis of

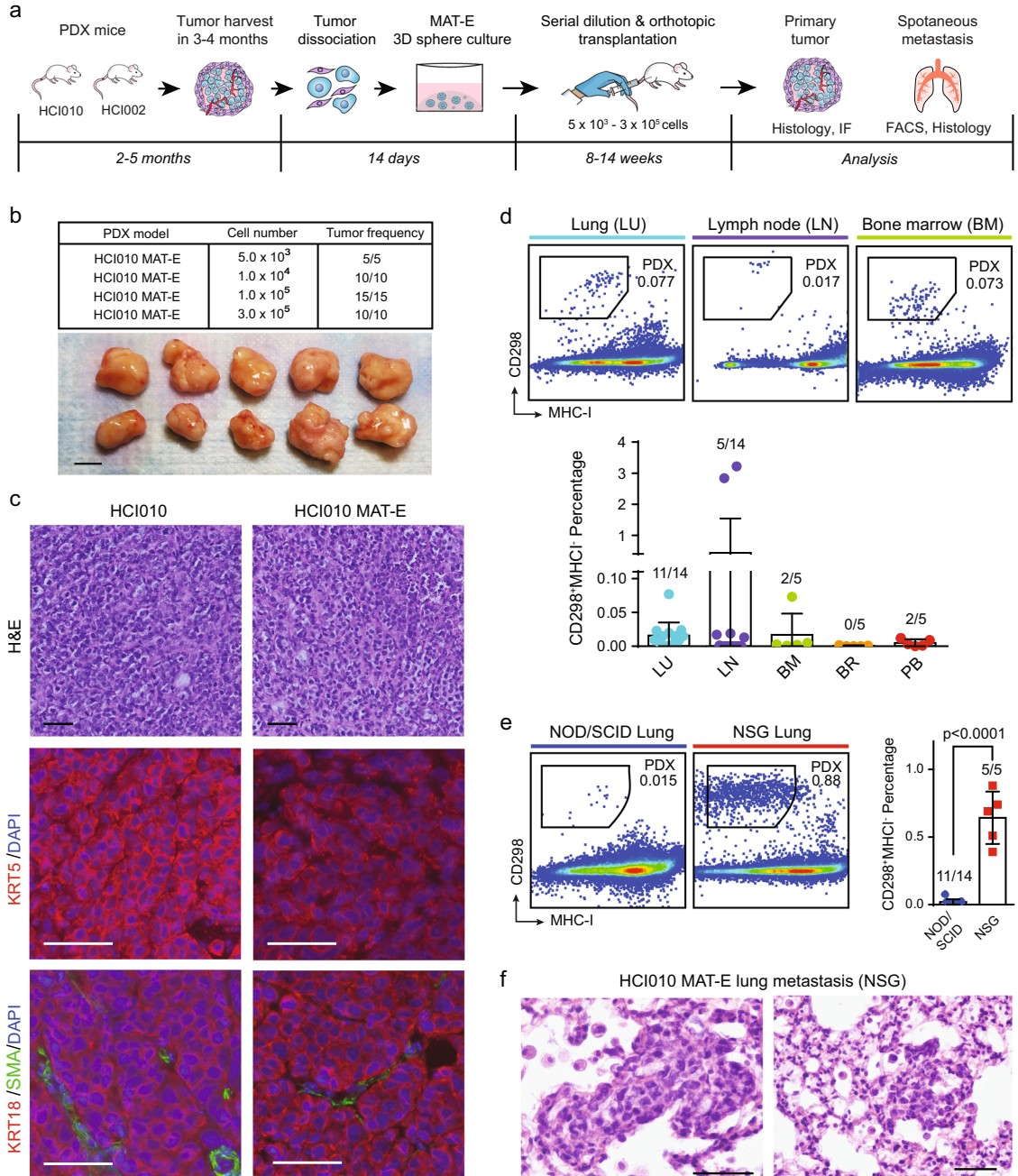

**Fig. 3 PDX tumor sphere cells maintain tumorigenic potential and form spontaneous metastasis in vivo. a** Schematic overview of workflow and timeline for assessing the capacity of cultured PDX cells to generate primary tumors and spontaneous metastasis following orthotopic transplantation. **b** Serial dilution and transplantation analysis to determine the tumorigenic capacity of cultured cells. HCI010 cells were cultured in MAT-E conditions and injected orthotopically into NOD/SCID mice at increasing dilution ($5.0 \times 10^3 - 3.0 \times 10^5$). Table (top) shows the fraction of tumors generated at each dilution. Representative images (bottom) show primary tumors generated from orthotopic transplantation of $1 \times 10^5$. Scale bar = 1 cm. **c** Histopathological and molecular marker analysis of tumors generated from uncultured (HCI010) and cultured (HCI010 MAT-E) cells. The top panels show the representative histopathological appearance of tumor sections stained with hematoxylin and eosin (H&E). Scale bar = 50 μm. The middle and bottom panels show representative images of IF staining for basal (KRT5), luminal (KRT18), and myoepithelial (SMA) cell markers. Scale bar = 50 μm. **d** Quantification of spontaneous metastasis in animals transplanted with $1 \times 10^5$ cultured HCI010 cells. Representative plots (top) show quantification of CD298+MHC-I− human metastatic PDX cells in distal tissues by flow cytometry. Bar graph (bottom) shows quantification of the percent of metastatic cells in a cohort of transplanted animals (n = 5–14). Fractions indicate the number of tissues with metastasis, defined by >0.005% CD298+MHC-I− cells. Data are represented as the mean ± s.d. *LU* lung, *LN* lymph node, *BM* bone marrow, *BR* brain, *PB* peripheral blood. **e** Comparison of lung metastatic burden in NOD/SCID and NSG animals transplanted orthotopically with $1 \times 10^5$ cultured HCI010 cells. Representative plots show CD298+MHC-I− human metastatic PDX cells in the lungs 8 weeks after transplant by flow cytometry. The bar graph shows the quantification of the percentage of metastatic cells in a cohort of transplanted animals. *P* value determined by unpaired *t* test. Data are represented as the mean ± s.d. Fractions indicate the number of tissues with metastasis, defined by >0.005% CD298+MHC-I− cells. **f** Representative images of metastatic lesions in the lungs of NSG animals transplanted with cultured HCI010 cells identified by H&E staining and histopathological analysis. Scale bar = 50 μm.

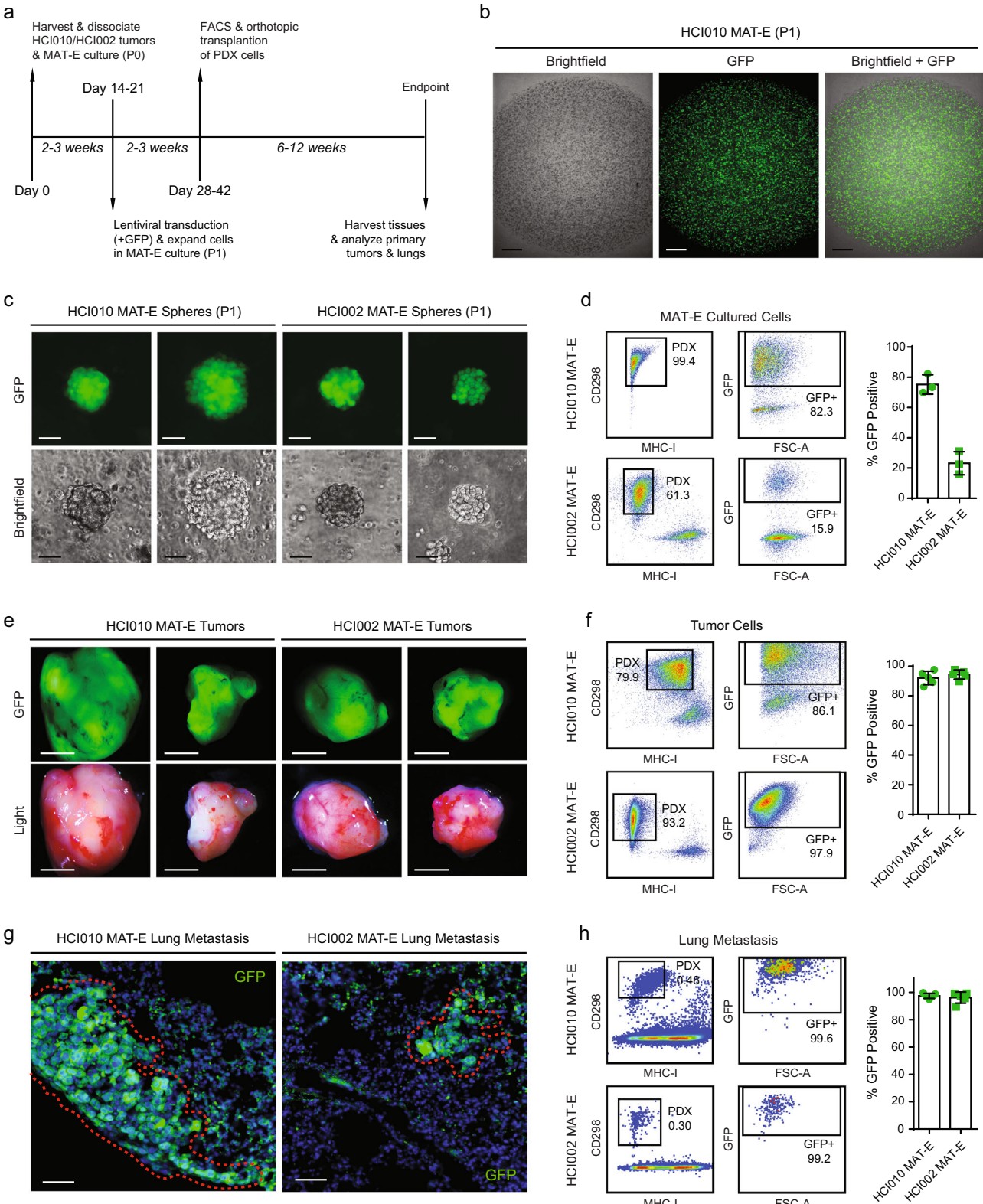

HCI002, as NSG mice supported more robust spontaneous metastasis (Fig. 3e). Indeed, flow cytometry analysis showed 5/5 NSG mice developed lung metastasis, compared with 0/5 NOD/ SCID mice (Supplementary Fig. 3c). Histopathological analysis confirmed the presence of metastatic lesions in the lungs, and IF staining showed specific expression of the basal cancer subtype marker KRT5 (Fig. 5f, g, Supplementary Fig. 3d). These data demonstrate that patient tumor cells can be expanded in culture and used in experimental metastasis assays.

**Validation of PDX culture-transplant system for authentic investigations of metastasis in vivo.** To investigate whether our culture-transplant system can be used to faithfully study

**Fig. 4 PDX tumor sphere cells can be genetically engineered for functional studies of metastasis in vivo. a** Schematic shows workflow for lentiviral engineering of PDX sphere cells in vitro followed by an analysis of their capacity to form primary tumors and spontaneous metastasis following transplantation in vivo. **b** Representative images show GFP expression in PDX sphere cultures following lentiviral transduction in vitro. HCI010 P0 sphere cells were transduced with GFP lentivirus at MOI = 25 and seeded in MAT-E culture at a density of $2 \times 10^5$ cells per well. Images show z-stack micrographs of individual wells containing P1 spheres 1 week later. Scale bar = 800 μm. **c** Images show GFP expression in representative P1 PDX spheres two weeks post transduction in MAT-E culture. Scale bar= 50 μm. **d** Representative flow cytometry plots show transduction efficiency of P1 PDX sphere cells two weeks post transduction and growth in MAT-E culture. The bar graph quantifies the percentage of PDX cells positive for GFP in each well ($n = 3$). Values represent the mean ± s.d. **e** Images show representative PDX tumors produced 60 days post orthotopic transplantation. $1 \times 10^5$ CD298$^+$MHC-I$^-$GFP$^+$ P1 PDX cells were sorted by flow cytometry and transplanted into each mouse ($n = 5$). Scale bar= 5 mm. **f** Representative flow cytometry plots show the percent of human CD298$^+$MHC-I$^-$ human tumor cells that retain GFP expression following tumor growth in vivo. The bar graph quantifies the percentage of human cells positive for GFP in each tumor ($n = 5$). Values are represented as the mean ± s.d. **g** Images show representative GFP+ metastatic lesions in the lungs of transplanted animals from **e**. A dashed red outline highlights the perimeter of the lesions. Scale bar = 50 μm. **h** Representative flow cytometry plots show the percent of human CD298$^+$MHC-I$^-$ human metastatic cells in the lung that retain GFP expression. The bar graph quantifies the percentage of human cells positive for GFP in each lung ($n = 5$). Values are represented as the mean ± s.d.

metastasis, we tested whether known pharmacological inhibitors of metastasis have a similar effect on PDX cells post culture. In prior work, we found that mitochondrial oxidative phosphorylation (OXPHOS) is upregulated in spontaneous lung metastases in PDX mice, and that inhibition of OXPHOS with the complex V inhibitor oligomycin (oligo) attenuates lung metastasis[17]. However, functional studies were performed using surrogate cell line models, since functional experiments could not be performed with PDX cells. Here, we determined whether OXPHOS is also critical for the metastatic spread of patient tumor cells using our culture-transplant system (Fig. 6a).

We first confirmed that oligo treatment does not affect PDX cell viability. HCI002 cells were grown in MAT-E culture for 2 weeks (P0), treated with oligo for 72 hours, and assessed for viability by trypan blue exclusion and aV and PI staining. In contrast to treatment with cytotoxic agents, we observed no overt changes in PDX sphere morphology following oligo treatment (Supplementary Fig. 4a). Trypan blue exclusion analysis showed there was no significant difference in the percentage of live cells in control (72.3 ± 3.1%) and oligo (72.3 ± 10.2%) treated conditions ($p = 0.999$) (Supplementary Fig. 4b). Analysis of aV and PI by flow cytometry further showed no difference in the percentage of live cells (oligo, 77.8 ± 1.9%; control, 81.0 ± 3.1%) ($p = 0.206$) (Supplementary Fig. 4c, d).

We next confirmed that oligo treatment inhibits OXPHOS in PDX cells using fluorescence lifetime (FLIM) imaging. Previous work has shown that oligo induces cells to shift from OXPHOS to glycolysis for ATP production[17,36,37]. This shift can be observed using FLIM imaging of NADH, as the FLIM of NADH is longer when bound to enzymes involved in OXPHOS (3.4 ns) than when free-floating in the cytoplasm during glycolysis (0.4 ns)[38,39]. We treated HCI002 spheres with oligo for 6 hours and washed and replaced the media with fresh drug-free media (Fig. 6b). FLIM showed a 7.4 ± 0.8% increase in the fraction of free NADH at 0 hours ($p < 0.0001$), which was sustained and slightly increased at 24 hours (8.9 ± 0.1%) ($p < 0.0001$) (Fig. 6b), confirming that oligo induces a shift from OXPHOS to glycolysis in cultured PDX cells.

We determined whether OXPHOS inhibition suppresses the metastatic capacity of cultured PDX cells following i.v. injection. HCI002 P0 sphere cells were treated with oligo or vehicle control and injected i.v. (Fig. 6a). Lungs were harvested eight weeks later and the percentage of human CD298$^+$MHC-I$^-$ metastatic cells in each condition was compared by flow cytometry. This revealed a 5.9-fold lower percentage of metastatic cells in mice injected with oligo (0.02 ± 0.01%, $n = 5$) compared with vehicle (0.12 ± 0.09%, $n = 6$) treated cells ($p = 0.03$) (Fig. 6c). These data validate our culture-transplant system for authentic investigations of

metastasis and show that OXPHOS is critical for metastasis of actual patient tumor cells.

**NME1 promotes lung metastasis from patient tumor cells**. We next used our culture-transplant system to investigate potential mechanisms underlying OXPHOS-mediated lung metastasis in PDX animals. In prior work, we found that NME1 is upregulated in spontaneous lung metastases that display increased OXPHOS (Fig. 7b)[17]. Increased NME1 expression in primary breast tumors is also associated with a worse prognosis in breast cancer (Supplementary Fig. 5a). NME1 catalyzes the transfer of a phosphate from nucleoside triphosphates to nucleoside diphosphates, primarily from ATP to GDP to produce GTP for use in G protein signaling[40]. Several G proteins such as RAC1 and CDC42 promote cell migration and growth and have established pro-metastatic functions[41]. We hypothesized that OXPHOS promotes metastasis in PDX cells through NME1, and tested whether NME1 overexpression promotes metastasis using our culture-transplant system.

HCI010 cells were cultured in MAT-E conditions for 2 weeks (P0) and transduced with lentivirus to overexpress NME1 and GFP (+NME1-GFP) or GFP alone (+GFP) (Fig. 7a). Transduced P0 cells were re-plated and expanded in culture for 3 weeks (P1) (Supplementary Fig. 5b), and human GFP$^+$ CD298$^+$MHC-I$^-$ PDX cells were isolated by flow cytometry and transplanted orthotopically into recipient mice for 60 days (Supplementary Fig. 5c). qPCR analysis for NME1 showed 3.9-fold increased expression in +NME1-GFP compared with +GFP transduced cells ($p = 0.0005$), confirming successful overexpression (Supplementary Fig. 5d). Analysis of primary tumor growth showed no difference in the growth kinetics or mass of +NME1-GFP ($n = 6$) and +GFP ($n = 6$) groups (Fig. 7c). However, whole-mount fluorescence microscopy analysis of lung tissues showed substantially more GFP signal in the +NME1-GFP group, indicating a higher metastatic burden (Fig. 7d). This was confirmed by flow cytometry, which showed a greater than fourfold increase in the percentage of GFP$^+$CD298$^+$MHC-I$^-$ human metastatic cells in the lungs of +NME1-GFP (0.98 ± 0.25%) compared with +GFP (0.23 ± 0.15%) control mice (Fig. 7e). These findings reveal a pro-metastatic function for NME1 in breast cancer lung metastasis and provide proof-of-principle for the value of our PDX culture-transplant system for functional investigations of new genes in metastasis using patient tumor cells.

## Discussion
Despite improvements in outcomes for early-stage patients, survival rates for metastatic patients have shown limited improvement and

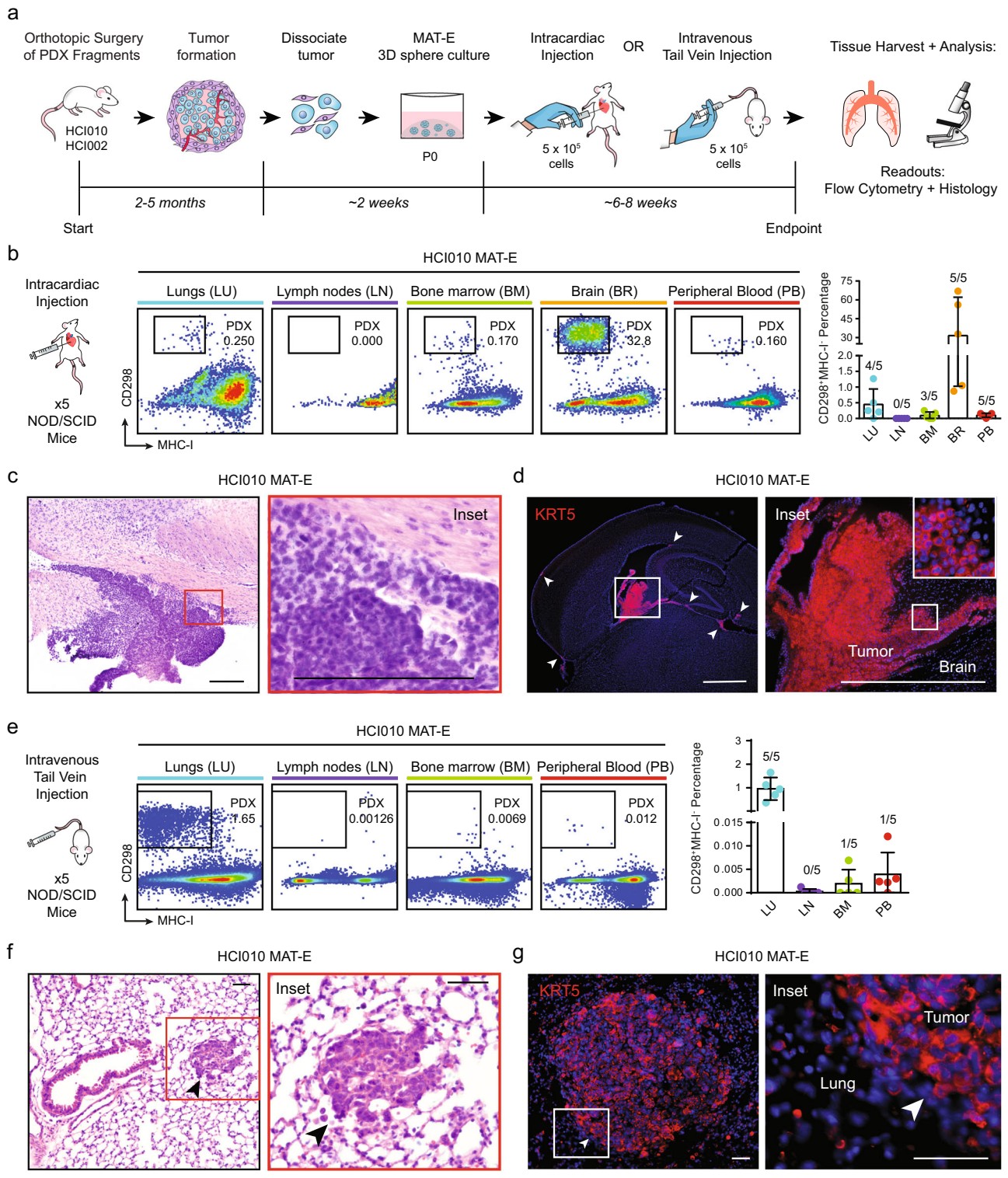

metastasis continues to be the major cause of death for cancer patients. A significant roadblock to progress has been the limited availability of models that faithfully recapitulate the biological complexities of metastasis in patients. Despite recent efforts to build large banks of human PDX models to improve authenticity in cancer research, their application in metastasis has been limited by the technical challenges of culturing patient cells in vitro for subsequent functional studies[42,43]. Here, we develop a robust approach for the viable propagation and engineering of PDX cells in 3D culture for functional studies of metastasis. Using orthotopic

transplantation and RNA sequencing, we show that cultured cells maintain tumorigenic potential and display minimal transcriptome changes, indicating that our method facilitates an authentic long-term culture of patient tumor cells. We show PDX cells engineered in vitro display robust capacity for spontaneous and experimental metastasis, demonstrating the value of our culture-transplant system for functional studies of metastasis in diverse contexts. To our knowledge, this represents the first adaptation of PDX models for metastasis research in vivo, presenting an important technical advance that may help drive innovation for the development of

**Fig. 5 PDX tumor sphere cells produce robust experimental metastasis in vivo. a** Schematic overview of workflow and timeline for assessing the capacity of cultured PDX cells to metastasize using i.c and i.v. experimental metastasis models. **b** Analysis of metastatic spread following i.c. injection of cultured PDX cells in vivo. In all, $5 \times 10^5$ cultured HCI010 cells were injected i.c. into NOD/SCID animals and organs were harvested eight weeks later. Flow cytometry plots show human CD298+MHC-I− human metastatic cells in the lungs, lymph nodes, bone marrow, peripheral brain, and blood of representative animals (left panels). The bar graph shows quantification of the percentage of metastatic cells in a cohort of transplanted animals ($n = 5$) (right panel). Fractions indicate the number of tissues with metastasis, defined by >0.005% CD298+MHC-I− cells. Data are represented as the mean ± s.d. **c** Images show representative metastatic lesion identified by histopathological analysis of H&E stained brain sections (standard 40 µm) from animals described in **b**. Inset shows higher power visualization of densely packed tumor cell nuclei within the lesion. Scale bar = 100 µm. **d** IF staining for KRT5 in representative brain section from animals described in **b**. Inset shows higher power visualization of KRT5 staining. Arrows indicate KRT5+ micrometastatic lesions. Scale bar = 800 µm. **e** Analysis of metastatic spread following i.v. injection of cultured PDX cells in vivo. In all, $5 \times 10^5$ cultured HCI010 cells were injected i.v. into NOD/SCID animals and organs were harvested eight weeks later. Flow cytometry plots show human CD298+MHC-I− human metastatic cells in the lungs, lymph nodes, bone marrow, and peripheral blood of representative animals (left panels). The bar graph shows the quantification of the percent of metastatic cells in a cohort of transplanted animals ($n = 5$) (right panel). Fractions indicate the number of tissues with metastasis, defined by >0.005% CD298+MHC-I− cells. Data are represented as the mean ± s.d. **f** Images show representative metastatic lesions identified by histopathological analysis of H&E stained sections of lung tissue from animals described in **c**. Arrows indicate metastatic lesions. Scale bar = 50 µm. **g** Images show representative IF staining for KRT5 in lung metastatic lesions from animals described in **c**. Arrows indicate metastatic lesions. Scale bar = 50 µm.

more efficacious therapeutic strategies tailored specifically for metastatic disease.

In addition to technical advances, our study highlights the critical importance of cellular metabolism in breast cancer metastasis. Despite the historical focus on Warburg metabolism in tumor biology, we and others have recently shown that mitochondrial OXPHOS is critical for metastatic progression. However, these studies were conducted using cell line models[17,44,45], so it remained unclear to what degree OXPHOS is important for metastasis in breast cancer patients. Here, we used our culture-transplant system to find that OXPHOS inhibition significantly attenuates the lung metastatic capacity of patient tumor cells. We further investigated the potential role of the metabolic enzyme NME1 in OXPHOS-mediated metastasis[17]. Although prior work in melanoma established NME1 as a metastasis suppressor gene[46,47], others have reported metastasis-promoting functions of NME1 in other cancers including neuroblastoma[48]. Similarly, we recently reported its upregulation in breast cancer lung metastases that display increased OXPHOS[17]. Given that the main function of NME1 is to transfer a phosphate from ATP to produce GTP, we hypothesized that NME1 may use ATP generated through OXPHOS to create GTP for G protein signaling pathways that promote metastasis[40,41]. Using our culture-transplant system, we found that NME1 overexpression results in a fourfold increase in spontaneous lung metastasis. These results reveal a new pro-metastatic role for NME1 in breast cancer, suggesting its function is tissue-context specific, and open the door to future studies investigating the mechanistic link between NME1 and OXPHOS and their potential as therapeutic targets for metastatic disease. Our study also provides key proof-of-principle for the value of our PDX culture-transplant system for functional investigations of new genes in metastasis using patient tumor cells.

## Methods
**Mouse strains**. NOD/SCID and NSG mice were purchased from The Jackson Laboratory (Bar Harbor, Maine, USA). All mice were maintained in a pathogen-free facility. All mouse procedures were approved by the University of California, Irvine, Institutional Animal Care, and Use Committee. Female mice were used between 4 and 6 weeks of age.

**Harvesting and processing tumors into single cells**. After 3–6 months of growth, tumors were collected from mice and mechanically dissociated, followed by 2 mg/mL collagenase (Sigma-Aldrich, Cat. no. C5138-1G) digestion in medium (Dulbecco's Modified Eagle Medium (DMEM) F-12 medium with 5% fetal bovine serum (FBS) at 37 °C for 45 minutes on a standard shaker. The digested tumor was washed with PBS, incubated with trypsin (Corning, Cat. no. 25-052-Cl) for

10 minutes at 37 °C, washed again, and then treated with DNaseI (Worthington Biochemical, Cat. no. LS002139). The cells were filtered through a 100 µm strainer. Live-cell concentration was checked using the Countess II automated cell counter (ThermoFisher Scientific Inc., Carlsbad, CA, USA).

**Culturing PDX cells**. PDX tumors were dissociated into single cells and plated in various culture conditions. PDX cells were plated in six-well ULA plates (Fisher Scientific, Cat. no. 07-200-601) at $2.5 \times 10^5$ cells/well and topped with 1 mL of either EpiCult™ Medium (StemCell Technologies, Cat. no. 05610) supplemented with 5% FBS, 10 ng/mL human epidermal growth factor, and 10 ng/mL basic fibroblast growth factor or MEGM™ culture medium (Lonza, Cat. no. CC-3150), completed as per manufacturer's protocol. The medium was changed when necessary.

PDX single cells were also cultured in standard flat-bottom 24-well plates. The cells were plated at a seeding density of $1.0 \times 10^5$ to $2.5 \times 10^5$ cells/well in a 1:1 mix of matrigel (growth factor reduced) (Corning, Cat. no. 356231) and EpiCult™ or MEGM™. In all, 1 mL of EpiCult medium was added to each well and changed as necessary. Cells were harvested anytime between days 9–21 using dispase (StemCell Technologies, Cat. no. 07913) and trypsin to dissociate matrigel and spheres. All PDX cells from all culture conditions were grown at 37 °C and 5% CO₂.

**Generation of RNA-sequencing data set**. We generated a data set of three biological replicates of paired HCI010 uncultured and cultured cells and paired HCI002 uncultured and cultured cells. We harvested and dissociated HCI010 and HCI002 PDX tumors using the methods described above. We split our single-cell suspension in half and stained half of the uncultured HCI010 and HCI002 cells for FACS using fluorescent-labeled antibodies for human-specific CD298 (Biolegend, Cat. no. 341704) and mouse-specific MHC-I (eBioscience, Cat. no. 17-5957-82). In all, $1.0 \times 10^5$ uncultured cells were sorted for each patient for RNA isolation. The other half of the cell suspension was used for culturing. HCI010 and HCI002 PDX cells were plated in 24-well plates at a seeding density of $2.5 \times 10^5$ cells/well and topped off with 1 mL of EpiCult. Cells were grown in culture for 2 weeks, and then harvested using dispase and trypsin. Cultured cells were stained for FACS using fluorescent-labeled antibodies for human-specific CD298 and mouse-specific MHC-I. In all, $1.0 \times 10^5$ cultured cells were sorted for each patient. After sorting, uncultured and cultured cells were processed for RNA extraction, followed by cDNA synthesis and amplification. Library construction was performed with the Takara Clontech SMARTer Stranded Total RNA-seq kit v2. The libraries were sequenced at a depth of 35 million paired-end reads for each sample on the NovaSeq 6000.

**Processing and analysis of bulk RNA-seq data**. Quality of reads was accessed using FastQC software. Reads were aligned to reference genome GRCh38/hg38 from UCSC using Spliced Transcripts Alignment to a Reference (STAR) software. The un-normalized count matrix was filtered to exclude any genes with 0 reads, and we designated the uncultured samples as the reference level. We then performed differential gene expression analysis using DESeq2 1.28.1 in R 4.0.0.

**Orthotopic transplantation and i.c and i.v injection models of cultured PDX cells for modeling metastasis**. PDX cells were harvested after 2–3 weeks of culture in matrigel and EpiCult. Dispase and trypsin were used to dissociate matrigel and spheres into single cells. For orthotopic transplants, $2.0 \times 10^4$–$1.0 \times 10^5$ cells were suspended in 15–20 µL of a 1:1 mixture of PBS and matrigel and injected into the number 4 mammary fat pads of NOD/SCID and

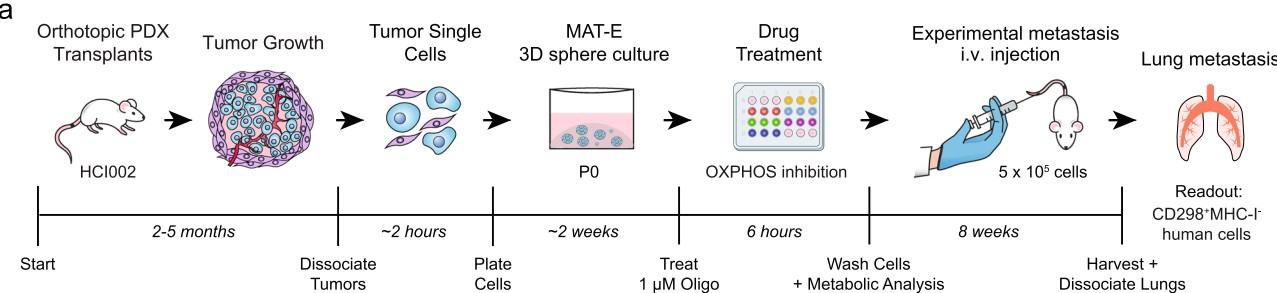

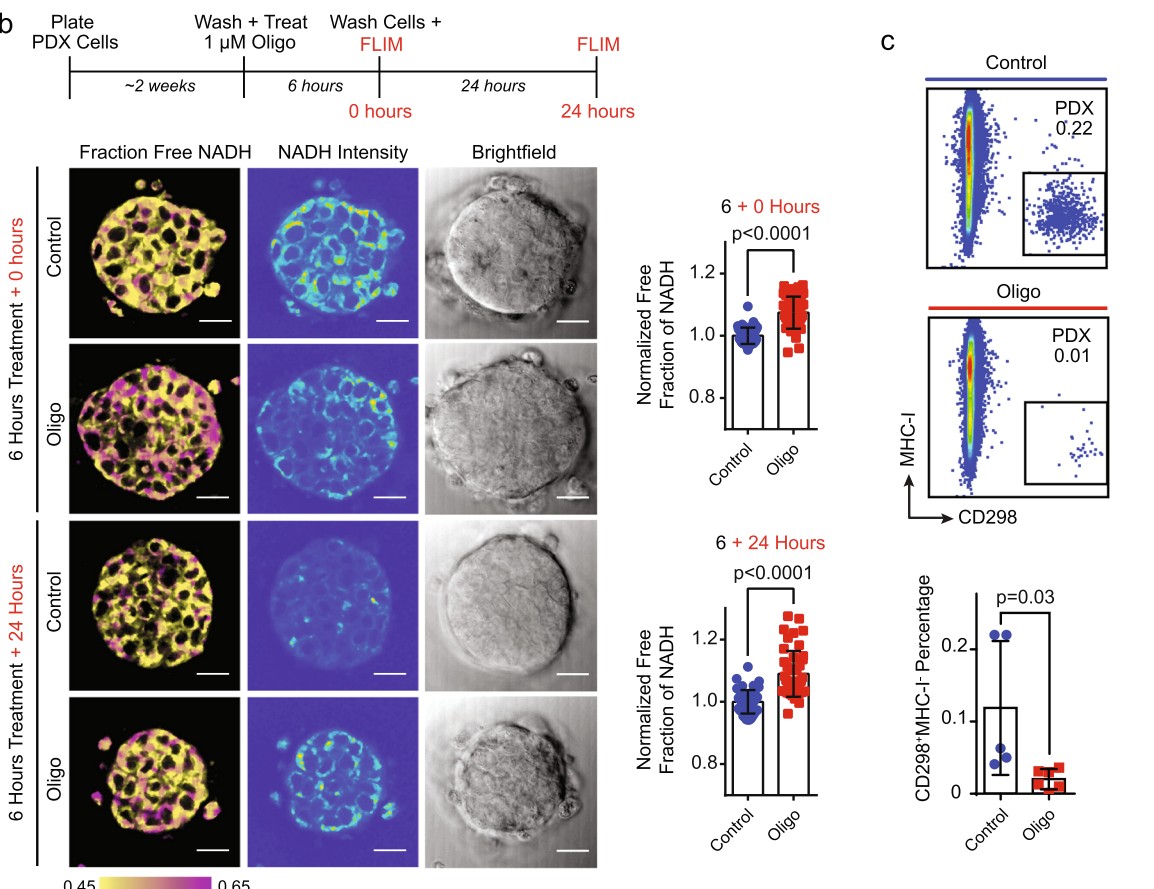

**Fig. 6 Validation of PDX culture-transplant system for authentic investigation of metastasis in vivo. a** Schematic shows experimental workflow to test the effects of OXPHOS inhibition on the metastatic capacity of cultured HCI002 PDX cells following i.v. injection in vivo. **b** Evaluation of the effects of oligo treatment on OXPHOS in PDX tumor spheres by FLIM imaging. Schematic shows experimental workflow (top panel). FLIM images show NADH fluorescence signal in representative HCI002 spheres immediately after treatment with 1 μM oligo or vehicle and replacement with drug-free medium (time = 0 hours) and 24 hours later (+24 hours) (bottom left panels). Scale bar = 20 μm. Bar graphs show quantification of unbound free NADH fraction at 0 and 24 hours (right panels). Values are normalized to the free NADH fraction in control cells at each time point for each replicate, such that a value of 1 represents no change in free NADH compared with control. Values are presented as the mean ± s.d. P value determined by unpaired t test. Homoscedasticity was determined via F test. Control n = 47 spheres at time t = 0 and 24 hours; oligo n = 46 spheres at time t = 0 and 24 hours. **c** Comparison of metastatic burden in mice transplanted with oligo vs. vehicle-treated PDX cells. In all, 5 × 10⁵ HCI002 PDX sphere cells were treated with 1 μM oligo or vehicle for 6 hours, washed and transplanted i.v. as described in **a**. Flow cytometry plots show the percent of human CD29⁺MHC-I⁻ metastatic cells in the lungs of representative mice eight weeks later. The bar graph shows the quantification of the percent of metastatic cells in a cohort of control (n = 5) and oligo (n = 6) animals. Values are expressed as mean ± s.d. P value determined by unpaired t test.

NSG mice using a 25 μL Hamilton syringe. For experimental metastasis models, cells were suspended at a concentration of 5 × 10⁶ cells/mL. In all, 5 × 10⁵ cells in 100 μl of PBS were injected into the right ventricle of each NOD/SCID or NSG mouse for i.c. injections or into the tail vein for i.v. injections.

**Analysis of primary tumors and metastasis in distal tissues**. Following 8–14 weeks post orthotopic transplants, or ~8 weeks post i.c. or i.v. injections,

primary tumors and peripheral tissues including lungs, lymph nodes, bone marrow, peripheral blood, and brains were harvested, dissociated to single cells and stained with fluorescently conjugated antibodies for CD298 (Biolegend, Cat. no. 341704) and MHC-I (eBioscience, Cat. no. 17-5957-82), and flow cytometry was used to analyze and quantify disseminated CD298⁺MHC-I⁻ PDX cells using the BD FACSAria Fusion cell sorter (Becton, Dickinson and Company, Franklin Lakes, NJ, USA) as described previously[26]. Primary tumor volumes were calculated using caliper measurements with the equation: volume of an ellipsoid = 1/2(length ×

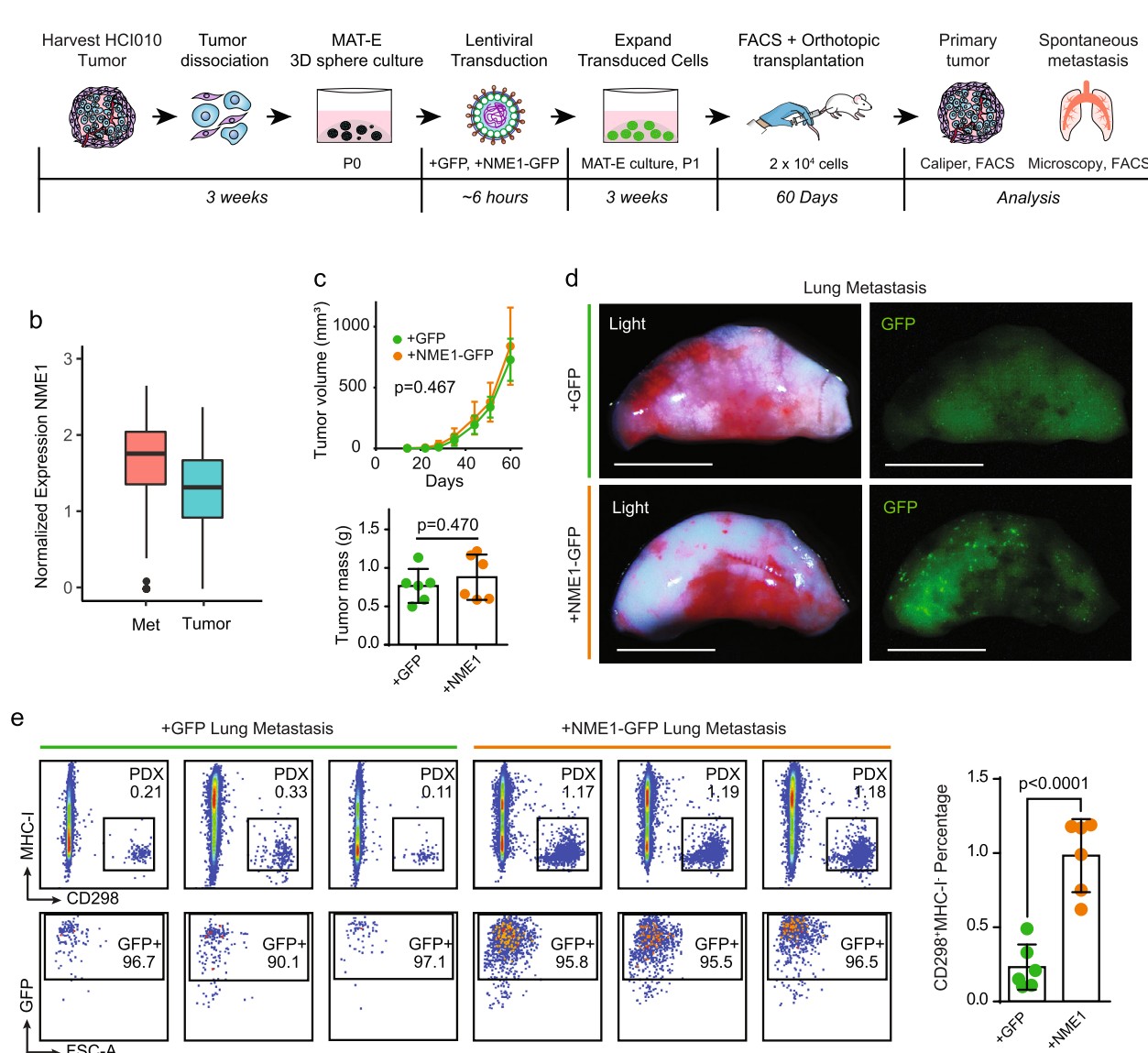

**Fig. 7 NME1 promotes lung metastasis from patient tumor cells. a** Schematic of experimental workflow to investigate the effect of NME1 overexpression on spontaneous metastasis from cultured PDX tumor cells. **b** The box plot shows normalized gene expression of NME1 in metastatic (red, $n = 435$ cells) and primary tumor cells (blue, $n = 684$ cells). The lower and upper hinges correspond to the first and third quartiles, and the midline represents the median. The upper and lower whiskers extend from the hinge up to 1.5 * IQR (inter-quartile range). Outlier points are indicated if they extend beyond this range. **c** Primary tumor weight and growth kinetics. $2 \times 10^4$ HCI010 PDX cells transduced with +NME1-GFP or +GFP lentivirus were transplanted orthotopically and monitored for 60 days ($n = 6$ each condition). The line graph shows tumor volume calculated by caliper measurements (top panel). The bar graph shows tumor weight measured at the endpoint (bottom panel). Values are expressed as mean ± s.d. *P* values determined by unpaired *t* test. **d** Images show GFP signal identifying metastatic nodules in representative lungs from orthotopically transplanted mice from **c**. Scale bar = 5 mm. **e** Comparison of lung metastatic burden generated from control versus NME1-overexpressing PDX cells. Flow cytometry plots show the percentage of human metastatic cells in the lungs of three representative mice per group, identified by CD298+MHC-I− (top left panels) and GFP (bottom left panels). The bar graph quantifies the frequencies of CD298+MHC-I− metastatic cells in the lungs of each animal in the cohort. Values are expressed as mean ± s.d. *P* value determined by unpaired *t* test.

width[2]). Whole-mount images of primary tumors and lungs were taken with a Leica MZ10 F modular stereomicroscope (Leica Microsystems, Buffalo Grove, IL, USA).

**Histology and pathological analysis**. Tissues were harvested from mice. The lungs and tumors were fixed in 4% formaldehyde. After overnight fixation at 4 °C, the tissues were processed for paraffin embedding using standard protocols. The paraffin-embedded tissues were cut into 7 μm sections using a Leica microtome, rehydrated, and stained with hematoxylin and eosin. For immunofluorescent staining, slides were subjected to antigen retrieval in a 10 mM citric acid buffer (0.05% Tween 20, pH 6.0). We then blocked non-specific binding with a blocking buffer (0.1% Tween 20 and 10% FBS in PBS) for 20 mins, and then incubated

tissues overnight at 4 °C with primary antibodies. Slides were washed with PBS, and then incubated at room temperature with secondary antibodies for an hour, followed by washing with PBS, and mounting with VECTASHIELD Antifade Mounting Medium with DAPI (Vector Laboratories, H-1200-10). Images were taken using a BZ-X700 Keyence microscope (Keyonce Corp. of America, Itasca, IL, USA). Pathological analysis was performed by Dr. Robert A. Edwards, a board-certified pathologist and the director of the Experimental Tissue Resource in the UC Irvine Health Chao Family Comprehensive Cancer Center.

**Lentivirus transduction for genetic engineering of PDX cells**. PDX cells were cultured for ~2–3 weeks in matrigel and EpiCult, removed from matrigel with dispase, dissociated to single cells with trypsin, neutralized with DMEM medium

with 10% FBS, and washed with PBS with a 5 min centrifugation at $300 \times g$. Cells were suspended in premade lentivirus solution at a multiplicity of infection of 25 and the final volume was topped off to no >100 μL with Opti-MEM$^{TM}$ I Reduced Serum Medium (ThermoFisher Scientific, Cat. no. 31985070). GFP control (+GFP) and NME1 (+NME1) lentiviral expression vectors were packaged into lentiviral particles and purchased from VectorBuilder Inc., Chicago, IL, USA (+GFP Cat. no. VB190812-1255tza, +NME1 Cat. no. VB180802-1083ueg). Cells were subjected to an hour-long spinfection at $300 \times g$ at 4 °C, resuspended, and incubated in the same lentivirus solution for an additional 4–5 hours in 96-well round-bottom ULA plate wells at 37 °C and 5% CO$_2$. Cells were centrifuged for 5 min at $300 \times g$, seeded in matrigel and EpiCult as described above, and expanded for an additional 2–3 weeks at 37 °C and 5% CO$_2$. The medium was replaced as needed. After 2–3 weeks in matrigel and EpiCult cultured, PDX cells were removed from matrigel with dispase, dissociated to single cells with trypsin to form single-cell suspensions for transplantation into mice. Cell sorting for GFP-positive CD298$^+$MHC-I$^-$ PDX cells with fluorescently tagged antibodies for CD298 (Biolegend, Cat. no. 341704) and MHC-I (eBioscience, Cat. no. 17-5957-82) was performed using the BD FACSAria Fusion cell sorter (Becton, Dickinson and Company, Franklin Lakes, NJ, USA).

**Pharmacologic studies.** Cells were treated with oligo (MP Biomedicals, Cat. no. 0215178610) for the inhibition of OXPHOS and treated with Taxol (Sigma-Aldrich Canada, Cat. no. T7402), staurosporine (STS) (Sigma-Aldrich Canada, Cat. no. S4400), and doxorubicin (Sigma-Aldrich Canada, Cat. no. D1515) as positive controls for cell death. All compounds were prepared as concentrated stock solutions in DMSO and cells were treated with diluted working solutions in a culture medium at DMSO concentrations no greater than 0.5% (v/v) using drug doses based on previous studies[17,49].

**Cell death and viability assays.** Trypan Blue exclusion assays were performed with the Countess II automated cell counter (ThermoFisher Scientific Inc., Carlsbad, CA, USA). For analysis of cell death, cells were stained with aV-FITC, diluted 1:100 (GeneTex Cat. no. GTX14082) and PI, diluted 1:100 (ThermoFisher Scientific Cat. no. P3566) for 15 mins in aV binding buffer (10 mM HEPES, 140 mM NaCl, 2.5 mM CaCl$_2$, pH 7.4). Fluorescence microscopy was performed with the BZ-X700 Keyence microscope (Keyonce Corp. of America, Itasca, IL, USA), and flow cytometry analysis was performed using the BD FACSAria Fusion cell sorter (Becton, Dickinson and Company, Franklin Lakes, NJ, USA).

**NADH FLIM imaging.** NADH FLIM lifetime images were acquired with an LSM 880 confocal microscope (Zeiss) with a $40 \times 1.2$ NA C-Apochromat water-immersion objective coupled to an A320 FastFLIM acquisition system (ISS). A Ti:Sapphire laser (Spectra-Physics Mai Tai) with an 80 MHz repetition rate was used for two-photon excitation at 740 nm. The excitation signal was separated from the emission signal by a 690 nm dichroic mirror. The NADH signal was passed through a 460/80 nm bandpass filter and collected with an external photomultiplier tube (H7522P-40, Hamamatsu). Cells were imaged within a stage-top incubator kept at 5% CO$_2$ and 37 °C. FLIM data were acquired and analyzed with the SimFCS 4 software developed at the Laboratory for Fluorescence Dynamics at UC Irvine. Calibration of the system was performed by acquiring FLIM images of coumarin 6 (~10 μM), which has a known lifetime of 2.4 ns in ethanol, to account for the instrument response function.

**Phasor FLIM NADH fractional analysis.** NADH assumes two main physical states, a closed configuration when free in solution, and an open configuration when bound to an enzyme[50]. These two physical states have differing lifetimes, 0.4 ns when in its free configuration, and 3.4 ns when in its bound configuration[51–53]. To quantify metabolic alterations, we performed a fractional analysis of NADH lifetime by calculating individual pixel positions on the phasor plot along the linear trajectory of purely free NADH lifetime (0.4 ns) and purely bound NADH lifetime (3.4 ns). We quantified the fraction of free NADH by calculating the distance of the center of mass of a spheroid's cytoplasmic NADH FLIM pixel distribution to the position of purely bound NADH divided by the distance between purely free NADH and purely bound NADH on the phasor plot. These segmentation and phasor analysis methods are described in detail elsewhere[54].

**Single-cell analysis.** To analyze the differences in the expression of NME1 between metastases and tumors in these models, we accessed our previously published data set on these PDX models, which included single-cell gene expression profiles of metastatic and primary tumor cells[17]. The expression matrices were log-transformed into log[transcripts per kilobase million + 1] matrices and loaded into the Seurat analysis package[55].

**qPCR analysis.** RNA was extracted by using Quick- RNA Microprep Kit (Zymo Research, R1050) following the manufacturer's protocol. RNA concentration and purity were measured with a Pearl nanospectrophotometer (Implen). Quantitative real-time PCR was conducted using PowerUp SYBR green master mix (Thermo

Fisher Scientific, A25742) and primer sequences were found in Harvard primer bank and designed from Integrated DNA Technologies; GAPDH forward primer 5′-CTCTCTGCTCCTCCTGTTCGAC-3′, GAPDH reverse primer 5′-TGAGCGA TGTGGCTCGGCT-3′; NME1 forward primer 5′-AAGGAGATCGGCTTGTGGT TT-3′, NME1 reverse primer 5′-CTGAGCACAGCTCGTGTAATC-3′. Gene expression was normalized to the GAPDH housekeeping gene. For relative gene expression, $2^{-\Delta\Delta Ct}$ values were used. The statistical significance of differences between groups was determined by an unpaired $t$ test using Prism 6 (GraphPad Software, Inc).

**Relapse-free survival analysis.** Kaplan–Meier (KM) survival curves were generated to perform relapse-free survival analysis on breast cancer patient primary tumor microarray data (of all breast cancer patients and individual subtypes of breast cancer) from the KM plotter database[56]. All KM plots are displayed using the "split patients by median" parameter.

**Statistics and reproducibility.** Statistical analyses were performed using Graph-Pad Prism software. All experiments were repeated at least three times with reproducible results, with biological and technical replicates as defined in each experiment.

**Reporting summary.** Further information on research design is available in the Nature Research Reporting Summary linked to this article.

## Data availability

The authors declare that all data supporting the findings of this study are available within the article and its supplementary information files or from the corresponding author upon reasonable request. RNA-sequencing data are available on the GEO database under the accession code GSE153887. Source data can be found in Supplementary Data 2.

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

## Acknowledgements

We thank Dr. Zena Werb at the University of California, San Francisco for thoughtful discussions and feedback on project overview and study design. We thank Armani Oganyan, Nathan Ryan James, Anh Thien Phung, Scott Nguyen, Lannhi Nguyen, and Jennifer Nguyen for their technical assistance and animal handling. We thank Dr. Alana Welm at the Huntsman Cancer Institute for generously providing PDX models. We thank the Genomics High Throughput Facility at the University of California, Irvine for conducting the bulk RNA sequencing of the PDX samples. We wish to acknowledge the support of the Chao Family Comprehensive Cancer Center Experimental Tissue Resource, supported by the National Cancer Institute of the National Institutes of Health under award number P30CA062203. This study was supported by funds from the National Cancer Institute (K22 CA190511 to D.A.L., 1R01CA234496, and 4R00CA181490 to K.K.), the American Cancer Society (134389-RSG-20-039-01-DDC to D.A.L., 132551-RSG-18-194-01-DDC to K.K.), the Chan-Zuckerberg Initiative (CZF2019-002432 to D.A.L. and K.K.), the National Institutes of Health (P41-GM103540 to M.A.D. and A.E.Y.T.L, T32CA009054 to R.T.D through matched university funds), the National Science Foundation (1847005 to M.A.D. and NSF GRFP DGE-1839285 to A.E.Y.T.L) and the V Foundation (V2019-019 to D.A.L.). H.A. was supported by the University of Hail, Hail, Saudi Arabia for the Ph.D. Fellowship. D.M. was supported by the Canadian Institutes of Health Research Postdoctoral Fellowship.

## Author contributions

D.A.L. and K.K. supervised research. D.A.L., K.K., D.M. and G.A.H. designed research. D.M., G.A.H., A.E.Y.T.L., H.A., K.B., K.R.D., M.R., J.W.W., R.T.D, K.T.E., A.L., M.Y.G.M., R.L., R.A.E., M.A.D., K.K. and D.A.L. performed research; G.A.H., K.B. and J.W.W. performed bioinformatic analyses; D.M., G.A.H., K.K. and D.A.L. wrote the paper manuscript, and all authors discussed the results and provided comments and feedback.

## Competing interests

The authors declare no competing interests.
