## [Peer Review File · Communications Biology]

Reviewers' comments:

Reviewer #1 (Remarks to the Author):

Beginning with a clear and concise abstract, the authors present a well-written manuscript with appropriately scaled conclusions. Ma and Hernandez, et al., utilizes the well-characterized HCI series of PDX samples to develop a PDX culture system for expansion and manipulation. The authors have described a culture technique that retains significant cell viability and inconsequential or non-detectable changes in the context of cultured vs. uncultured PDX material harvested from their murine avatars. Their PDX-culture method preserves tumorigenicity and primary tumor characteristics and retains metastatic capability (spontaneous and experimental) after lentiviral manipulation. The system is robust enough to allow in vitro expansion/manipulation, re-graft, and primary/metastatic tumor characterization in a variety of settings, illustrating proof of principle for targeted inquiries into the in vivo metastatic process. The described culture process and its implications for future research utility would be of great interest to those studying the underpinnings of metastasis, especially with the authors' validation of an approach that preserves the tumor characteristics of primary patient samples.

Although the idea of using PDX to investigate metastasis is not entirely novel, there is little to no literature concerning aspects of the in vitro 3D expansion and organoid manipulation that is followed by orthotopic or intraperitoneal/intracardiac experimental metastasis injections to facilitate hypothesis-driven, mechanistic characterization of the metastatic process. The data presentation and analysis are thorough, appropriate, and well-described. Data and figures are of high quality and logically organized; the experimental schematic figures were very convenient to have during figure evaluation. The extended data figures were very helpful, as they alleviated several questions or concerns that would have been raised without their inclusion. Statistical tests appear appropriate and any corrections lean towards being conservative as to not overcall significance or variance. The conclusions drawn and data interpretation are strong and appropriate. The robustness of NME1 promoting metastasis (fig. 7) could benefit from further elaboration and refinement due to the prevalence of literature stating the opposite (or the authors' suggestions of tissue-specific NME1 effects), but in my view is not necessary for the 'proof of principle' nature of the manuscript.

Minor suggested improvements for revision: 1) Figure 3b mentions a scale bar in the legend that is not in the picture. 2) There is a missing or incomplete reference insertion in the legend of Extended Fig. 5a (in the figure file, but not the text file)

Reviewer #3 (Remarks to the Author):

The manuscript by Ma et al. described an in vitro 3D culture method for expanding and genetically manipulating PDX tumor cells. They compared 2 types of commercial medium either in suspension culture or embedded in Matrigel. They found that Matrigel plus the epicult medium was superior in maintaining the viability of PDX cells and for expansion than the other conditions. They showed that cultured cells were capable of forming primary tumors and metastases. The authors further demonstrated that the PDX spheres can be transduced with lentiviral vectors and still maintain tumor/metastasis forming ability. In the end, they functionally validated a previously identified metastasis-promoting gene, NME1, by overexpressing NME1 in the PDX cells. Overall, this is an interesting study that provides a useful methodology for studying metastasis with PDXs. However, there are several issues that need to be addressed.

1. It's not clear how efficient is the MAT-E condition in expanding PDX cells, especially during long-term serial passage. At one point, the authors stated that over a 14-17 day culture, they achieved a 2-fold expansion of viable cells, which was modest. The authors should perform serial passage

experiments for several PDX lines to determine a) whether this method can sustain long-term expansion, and b) what's the expansion rate.

2. The authors only tested the tumorigenicity and metastatic potential of PDX cells cultured up to one passage. Do later passage cells still maintain the tumorigenic and metastatic potential? This is important for distinguishing this work from previous studies that have done PDX cell transduction and reimplantation using short-term culture.

3. How does MAT-E compare to other human cancer cell organoid culture conditions, such as the condition reported in Sachs et al 2018 (PMID 29224780).

4. How do the metastatic abilities compare between PDX tumors and PDX sphere derived tumors?

5. In Figure 5B, intracardiac injection seemed to produce a massive metastatic burden in the brain, over 50% of all live cells. This is an astounding amount of metastatic burden! They need to confirm this with histology to rule out unknown artifacts.

Response to Reviewers Comments

Reviewer #1 (Remarks to the Author):

Beginning with a clear and concise abstract, the authors present a well-written manuscript with appropriately scaled conclusions. Ma and Hernandez, et al., utilizes the well-characterized HCI series of PDX samples to develop a PDX culture system for expansion and manipulation. The authors have described a culture technique that retains significant cell viability and inconsequential or non-detectable changes in the context of cultured vs. uncultured PDX material harvested from their murine avatars. Their PDX-culture method preserves tumorigenicity and primary tumor characteristics and retains metastatic capability (spontaneous and experimental) after lentiviral manipulation. The system is robust enough to allow in vitro expansion/manipulation, re-graft, and primary/metastatic tumor characterization in a variety of settings, illustrating proof of principle for targeted inquiries into the in vivo metastatic process. The described culture process and its implications for future research utility would be of great interest to those studying the underpinnings of metastasis, especially with the authors' validation of an approach that preserves the tumor characteristics of primary patient samples.

Although the idea of using PDX to investigate metastasis is not entirely novel, there is little to no literature concerning aspects of the in vitro 3D expansion and organoid manipulation that is followed by orthotopic or intraperitoneal/intracardiac experimental metastasis injections to facilitate hypothesis-driven, mechanistic characterization of the metastatic process. The data presentation and analysis are thorough, appropriate, and well-described. Data and figures are of high quality and logically organized; the experimental schematic figures were very convenient to have during figure evaluation. The extended data figures were very helpful, as they alleviated several questions or concerns that would have been raised without their inclusion. Statistical tests appear appropriate and any corrections lean towards being conservative as to not overcall significance or variance. The conclusions drawn and data interpretation are strong and appropriate. The robustness of NME1 promoting metastasis (fig. 7) could benefit from further elaboration and refinement due to the prevalence of literature stating the opposite (or the authors' suggestions of tissue-specific NME1 effects), but in my view is not necessary for the 'proof of principle' nature of the manuscript.

We thank for the reviewer for his/her thoughts and suggestions. We have added additional text to the Discussion section providing more information about the controversy over NME1 function in different cancer types. We note that prior work on NME1 showed an anti-metastatic role in melanoma (Stegg et al. 1988; Zhang et al. 2011) Consistent with this, Kaplan-Meier survival analysis has shown that high NME1 expression signatures are predictive of increased survival in melanoma patients (Leonard et al. 2018). Interestingly, controversy over its role has developed in recent years, where studies have begun to indicate an opposite, pro-metastatic role for NME1 in other cancers (Tan and Chang 2018). We provide sample Kaplan-Meier plots below showing that NME1 expression is predictive of poor prognosis in many cancer types in addition to breast cancer (Györfy et al. 2010). Our data (Fig 7) is consistent with a poor prognostic, pro-metastatic role for NME1 in triple negative breast cancer.

Trends in survivorship associated with NME1 expression in specific cancers. Left panel: Kaplan-Meier (KM) curve shows overall survival in melanoma patients based on their expression of an NME1 signature (taken from Leonard et al., 2018). Clusters 1-2 indicate low expression of the gene signature; Clusters 3-4 indicate high expression. **Right panels:** KM curves show relapse free survival (RFS) in patients based on their primary tumor expression of NME1, using the KM plotter database (Györfy et al. 2010). *P*-values were determined via a log-rank test.

Minor suggested improvements for revision: 1) Figure 3b mentions a scale bar in the legend that is not in the picture. 2) There is a missing or incomplete reference insertion in the legend of Extended Fig. 5a (in the figure file, but not the text file).

A scale bar has been added to **Figure 3b**, and **Extended Fig. 5a** (the figure file) has been revised to address the error.

Reviewer #3 (Remarks to the Author):

The manuscript by Ma et al. described an in vitro 3D culture method for expanding and genetically manipulating PDX tumor cells. They compared 2 types of commercial medium either in suspension culture or embedded in Matrigel. They found that Matrigel plus the epicult medium was superior in maintaining the viability of PDX cells and for expansion than the other conditions. They showed that cultured cells were capable of forming primary tumors and metastases. The authors further demonstrated that the PDX spheres can be transduced with lentiviral vectors and still maintain tumor/metastasis forming ability. In the end, they functionally validated a previously identified metastasis-promoting gene, NME1, by overexpressing NME1 in the PDX cells. Overall, this is an interesting study that provides a useful methodology for studying metastasis with PDXs. However, there are several issues that need to be addressed.

1. It's not clear how efficient is the MAT-E condition in expanding PDX cells, especially during long-term serial passage. At one point, the authors stated that over a 14-17 day culture, they achieved a 2-fold expansion of viable cells, which was modest. The authors should perform serial passage experiments for several PDX lines to determine a) whether this method can sustain long-term expansion, and b) what's the expansion rate.

We agree that long-term passaging is an important component of many culture methods. However, it was contrary to the primary goal of our particular application. The goal of our study was to establish conditions for in vitro propagation of human PDX models for use in functional metastasis assays. We deliberately refrain from long-term passaging of PDX cells to preserve their innate biology and distinguish them from human cell lines. We have added text to the manuscript to clarify this point.

We have also added new data to **Extended data Fig 1c** further evaluating the expansion rate in another PDX model. Similar to the HCI010 model, we observe 2.4-fold increase in cell number in HCI002 cells grown in MAT-E culture conditions. While this is modest, we show it is sufficient to propagate the cells for lentiviral engineering and functional studies in vivo. We also note that the 2-fold expansion is superior to what is achieved by other culture conditions, including those published by Sachs et al. (see below).

2. The authors only tested the tumorigenicity and metastatic potential of PDX cells cultured up to one passage. Do later passage cells still maintain the tumorigenic and metastatic potential? This is important for distinguishing this work from previous studies that have done PDX cell transduction and reimplantation using short-term culture.

Like above, long-term passaging was contrary to the goal of our study to generate a metastasis model. We have added text to the manuscript to clarify this point.

3. How does MAT-E compare to other human cancer cell organoid culture conditions, such as the condition reported in Sachs et al 2018 (PMID 29224780).

We compared four different culture conditions, some of which were adapted from prior reports on human organoid cultures. We compared two 3D culture methods and two media conditions. We tested MEGM media which has been used for transient cultures of PDX cells for drug testing (Bruna et al., Cell, 2016), and EpiCult-B media which has been used to culture normal and malignant human breast cells (Stingl et al., Breast Cancer Res Treat, 2001; Eirew et al., Nat Med, 2008; Diehn et al., Nature, 2009).

We also tested the conditions developed by Sachs et al., 2018. The authors developed an elaborate culture media to support the growth of organoids from primary patient tumor samples. We investigated whether this media could also support the growth of dissociated PDX tumor cells, and tested two different formats, 1) suspension (ULA plates) (**ULA-S**), and 2) Matrigel (**MAT-S**). Our results show that both conditions maintain viable PDX cells for >1 week (**see figure below**). However, while ULA-S yielded spheres, it did not produce cell expansion, and MAT-S resulted in few spheres and a net loss of viable cells (**figure below**). There may be several explanations for these results. It may be that xenografted breast tumor cells require different growth conditions than primary breast tumors, since they have already been established in mice. It is also possible that the Sachs conditions were designed for the growth of organoids, as opposed to the single cell suspensions utilized in our study. Finally, it is possible that the Sachs conditions do not support the growth of our specific patient models, which could be consistent with their report that it did not support growth of some patient samples.

Importantly, the main goal of our culture method is to sustain PDX tumor cells for use in metastasis assays in vivo, distinguishing it from prior work (Sachs et al and Bruna et al). Our culture method is also a simpler and utilizes more economical medium that would be more sustainable for the general research community, as the Sachs medium contains 16 supplements and costs >\$2000.

Analysis of culture conditions from Sachs *et al* 2018. (a) Representative brightfield images show sphere structures generated nine days after plating 1×10^5 HCI010 and HCI002 cells in MAT-S and ULA-S conditions. **(b)** Bar graph shows total viable HCI010 and HCI002 cell numbers nine days after plating 1×10^5 HCI002 cells (dashed line) by trypan blue exclusion. $n=6$ wells per condition. Data represented as mean \pm s.d.

4. How do the metastatic abilities compare between PDX tumors and PDX sphere derived tumors?

We have added text and data (**Extended Data Fig. 2c**) comparing metastatic frequencies from cultured PDX cells, to uncultured PDX cells from prior reports (Lawson *et al.*, 2015, *Nature*).

5. In Figure 5B, intracardiac injection seemed to produce a massive metastatic burden in the brain, over 50% of all live cells. This is an astounding amount of metastatic burden! They need to confirm this with histology to rule out unknown artifacts.

We have performed additional intracardiac injection experiments to evaluate brain metastatic burden by histology and immunofluorescence (IF) analysis. New data was added to **Figure 5c,d** and **Extended Data Fig 3a**. We found large metastatic lesions in the brains of injected animals. IF staining showed specific expression of the basal cancer subtype marker KRT5, as well as the proliferation marker Ki67, confirming their origin from PDX tumor cells. We have also provided additional images below, showing more examples of brain metastatic lesions in these animals (borders indicated by dashed lines).

We have also updated the text to note the range of metastatic burden observed in intracardiac injection experiments. All animals did not display massive burden (>50% of live cells). Some animals showed much less burden (1-5%), so we corrected the text to note this. Consistent with flow cytometry, our additional histological analysis showed the presence of more extensive lesions in some animals and less in others.

Brain metastatic lesions from PDX experimental metastasis model. 5×10^5 cultured HCl010 MAT-E cells were injected i.c. into NSG animals and brain tissue was analyzed seven weeks later. Images shown DAPI stained brain slices from 3 different animals. Insets show higher magnification of metastatic lesions. Dashed lines outline metastatic lesions.

References Cited:

- Calvo, Belén, Felipe Rubio, Miriam Fernández, and Pedro Tranque. 2020. "Dissociation of Neonatal and Adult Mice Brain for Simultaneous Analysis of Microglia, Astrocytes and Infiltrating Lymphocytes by Flow Cytometry." *IBRO Reports* 8. <https://doi.org/10.1016/j.ibror.2019.12.004>.
- Györfy, Balazs, Andras Lanczky, Aron C Eklund, Carsten Denkert, Jan Budczies, Qiyuan Li, and Zoltan Szallasi. 2010. "An Online Survival Analysis Tool to Rapidly Assess the Effect of 22,277 Genes on Breast Cancer Prognosis Using Microarray Data of 1,809 Patients." *Breast Cancer Research and Treatment* 123 (3): 725–31. <https://doi.org/10.1007/s10549-009-0674-9>.
- Leonard, M. Kathryn, Joseph R. McCorkle, Devin E. Snyder, Marian Novak, Qingbei Zhang, Amol C. Shetty, Anup A. Mahurkar, and David M. Kaetzel. 2018. "Identification of a Gene Expression Signature Associated with the Metastasis Suppressor Function of NME1: Prognostic Value in Human Melanoma." *Laboratory Investigation* 98 (3). <https://doi.org/10.1038/labinvest.2017.108>.
- Steeg, P S, G Bevilacqua, L Kopper, U P Thorgeirsson, J E Talmadge, L A Liotta, and M E Sobel. 1988. "Evidence for a Novel Gene Associated with Low Tumor Metastatic Potential." *Journal of the National Cancer Institute* 80 (3): 200–204. <https://doi.org/10.1093/jnci/80.3.200>.

- Tan, Choon Yee, and Christina L. Chang. 2018. "NDPKA Is Not Just a Metastasis Suppressor-Be Aware of Its Metastasis-Promoting Role in Neuroblastoma." *Laboratory Investigation*.
<https://doi.org/10.1038/labinvest.2017.105>.
- Zhang, Qingbei, Joseph R. McCorkle, Marian Novak, Mengmeng Yang, and David M. Kaetzel. 2011. "Metastasis Suppressor Function of NM23-H1 Requires Its 3'-5' Exonuclease Activity." *International Journal of Cancer* 128 (1): 40–50. <https://doi.org/10.1002/ijc.25307>.

REVIEWERS' COMMENTS:

Reviewer #1 (Remarks to the Author):

My comments and request for clarification from the initial review (Reviewer #1) have been addressed satisfactorily.

Reviewer #3 (Remarks to the Author):

I appreciate that the authors have addressed most of my original concerns. However, since their system has only been tested in a short-term culture and yielded limited increase of cell numbers, they should remove the claim of expansion, at least in the Abstract. What they described is really the maintenance of the cells. The term of expansion in organoid culture is associated with the exponential increase of cell numbers as shown in the classical intestinal organoids.

Point by Point Response to Reviewers' Comments

REVIEWERS' COMMENTS:

Reviewer #1 (Remarks to the Author):

My comments and request for clarification from the initial review (Reviewer #1) have been addressed satisfactorily.

Response: Thank you for your comments.

Reviewer #3 (Remarks to the Author):

I appreciate that the authors have addressed most of my original concerns. However, since their system has only been tested in a short-term culture and yielded limited increase of cell numbers, they should remove the claim of expansion, at least in the Abstract. What they described is really the maintenance of the cells. The term of expansion in organoid culture is associated with the exponential increase of cell numbers as shown in the classical intestinal organoids.

Response: Thank you for your comments. The notion of expansion has been removed in the abstract and throughout the manuscript.